# Pediatric COVID-19 patients in South Brazil show abundant viral mRNA and strong specific anti-viral responses

Tiago Fazolo [1,10], Karina Lima [1,10], Julia C. Fontoura [1,10], Priscila Oliveira de Souza [1], Gabriel Hilario [1], Renata Zorzetto [1], Luiz Rodrigues Júnior[1], Veridiane Maria Pscheidt[1], Jayme de Castilhos Ferreira Neto[1], Alisson F. Haubert[1], Izza Gambin[1], Aline C. Oliveira[1], Raissa S. Mello[1], Matheus de Bastos Balbe e Gutierres[1], Rodrigo Benedetti Gassen[2], Lais Durço Coimbra[3], Alexandre Borin [3], Rafael Elias Marques [3], Ivaine Tais Sauthier Sartor[4], Gabriela Oliveira Zavaglia [4], Ingrid Rodrigues Fernandes [4], Helder I. Nakaya [5], Fernanda Hammes Varela [4,6], Márcia Polese-Bonatto[4], Thiago J. Borges[2], Sidia Maria Callegari-Jacques [7], Marcela Santos Correa da Costa [8], Jaqueline de Araujo Schwartz [8], Marcelo Comerlato Scotta[4,6], Renato T. Stein[4,6] & Cristina Bonorino [1,9✉]

COVID-19 manifests as a milder disease in children than adults, but the underlying mechanisms are not fully characterized. Here we assess the difference in cellular or humoral immune responses of pediatric and adult COVID-19 patients to see if these factors contribute to the severity dichotomy. Children's non-specific immune profile is dominated by naive lymphocytes and HLA-DR$^{high}$CX3CR1$^{low}$ dendritic cells; meanwhile, children show strong specific antibody and T cell responses for viral structural proteins, with their T cell responses differing from adults by having weaker CD8$^+$TNF$^+$ T cells responses to S peptide pool but stronger responses to N and M peptide pools. Finally, viral mRNA is more abundant in pediatric patients. Our data thus support a scenario in which SARS-CoV-2 infected children contribute to transmission yet are less susceptible to COVID-19 symptoms due to strong and differential responses to the virus.

[1] Departamento de Ciências Básicas da Saúde, Universidade Federal de Ciências da Saúde de Porto Alegre - UFCSPA, Porto Alegre, Rio Grande do Sul, Brazil. [2] Center for Transplantation Sciences, Department of Surgery, Massachusetts General Hospital, Harvard Medical School, Boston, MA, USA. [3] Brazilian Biosciences National Laboratory, Brazilian Center for Research in Energy and Materials (CNPEM), Campinas, Sao Paulo, Brazil. [4] Social Responsibility – PROADI-SUS, Hospital Moinhos de Vento, Porto Alegre, Rio Grande do Sul, Brazil. [5] Hospital Israelita Albert Einstein, Jardim Leonor, Sao Paulo, Brazil. [6] Escola de Medicina, Pontifícia Universidade Católica do Rio Grande do Sul - PUCRS, Porto Alegre, Rio Grande do Sul, Brazil. [7] Departamento de Estatística, Universidade Federal do Rio Grande do Sul, Porto Alegre, Rio Grande do Sul, Brazil. [8] Coordenação-Geral do Programa Nacional de Imunizações, Departamento de Imunizações e doenças transmissíveis, Secretaria de vigilância em saúde - Ministério da Saúde (CGPNI/DEIDT/SVS/MS), Porto Alegre, Rio Grande do Sul, Brazil. [9] Department of Surgery, University of California at San Diego - UCSD, La Jolla, CA, USA. [10]These authors contributed equally: Tiago Fazolo, Karina Lima, Julia C. Fontoura. ✉email: cristinabonorino@gmail.com

Coronavirus disease 2019 (COVID-19) is a complex disease with multisystemic involvement, and an array of clinical manifestations that can vary from asymptomatic to severe outcomes leading to death[1] constituting an ongoing worldwide emergency[2]. Epidemiological evidence of less severe forms of the disease and reduced mortality in children upon infection with severe acute respiratory syndrome coronavirus 2 (SARS-CoV-2) is consistent[3,4], except for a multisystem inflammatory syndrome (MISC) associated with co-morbidities in a relatively low percentage of children[5]. The pediatric population (0–19 years old) represents more than 25% of the Brazilian population, however, it is observed that this group corresponds to only 1.9% (19,589/989,170) of all cases of COVID-19 reported in the past 12 months. Mortality (case fatality rate, or CFR—the proportion of deaths in identified confirmed cases), among children, represented 0.5% (1564/321,659) of all deaths due to the disease reported in the same period. The lethality in children and adolescents hospitalized due to SARS by COVID-19 was 8.0% (1574/19,589), while the overall lethality in all age groups was 32.5% (321,659/989,170), in the observed period (data from SIVEP-Gripe/Influenza Epidemiological Surveillance Information System, Brazilian Ministry of Health). Thus, a significantly lower number of children and adolescents have severe clinical presentations with the need for hospitalization, or that will lead to death when compared to other age groups.

Different hypotheses are used to explain this phenomenon[6,7]. Milder disease in children can result from a reduced expression of the viral receptor Angiotensin-converting enzyme 2 (ACE2), leading to lower levels of viral replication[8]. Alternatively, a differential immune response in children leads to a distinct infection course from adults;[9] or yet the pre-existence of neutralizing antibodies to seasonal coronaviruses could confer some cross-protection against SARS-CoV-2 induced disease. Children are considered one of the main reservoirs for these viruses[10], even though some studies show large circulation also among college students[11]. At present, the scarcity of data prevents a clear understanding of the striking differences between the pediatric and adult outcomes after infection by SARS-CoV-2.

Comprehensive studies have characterized immune responses in adults with mild or severe forms of COVID-19[12–15]. However, considerably fewer studies have focused on pediatric patients. This is a subject of paramount importance, not only because it is central to the design of public policies regulating school opening (and all the activities associated with it) during the pandemic, but also because understanding the milder disease presentation in children may provide important clues for the design of prevention strategies as well as novel therapeutic pathways for the management of COVID-19.

Here, we present a detailed characterization of plasma and peripheral blood mononuclear cells (PBMCs) from adult and pediatric COVID-19 patients by multi-parameter flow cytometry, defining 78 immune cell subsets. Using a systems approach, we analyze 38,670 data points, including anti-SARS-CoV-2 IgA and IgG antibodies, and frequencies of specific effector T cells. Taken together, our findings suggest that children produce a strong, yet differential immune response when compared to adults, which associates with the mild manifestation in pediatric COVID-19.

## Results

**Unsupervised analysis of non-specific immune responses in pediatric patients and adults with mild or severe disease.** The study design is summarized in Fig. 1A. We have recruited a total of 92 patients (25 children; 34 adults with mild disease—AMD; and 33 adults with severe disease—ASD). All subjects had COVID-19 confirmed by PCR detecting SARS-CoV-2 infection.

All children had mild disease and were treated as outpatients. Their characteristics are described in Table 1. The youngest individual enrolled was 7 months old—which does not appear in the table because only the interquartile interval (IQR) is shown. Most individuals were Caucasian. As expected, comorbidities were concentrated in the group with severe disease, which was also the group with a higher mean age. Some symptoms are probably not accurately assessed in some children, such as anosmia or dysgeusia, due to the age of some individuals in this group. Dyspnea was significantly less frequent in children. Median cycle threshold (Ct) levels for all three probes used in PCR were higher in AMD, and not different between ASD and children.

Comprehensive immune profiling of PBMCs from pediatric and adult patients was performed using flow cytometry, generating 77 variables (frequencies of cell subpopulations and geometric Mean Fluorescent Intensity/gMFI of activation markers). Gating strategies are detailed in Supplementary Fig. 1A–H). To reduce the dimensionality of the numerous variables, a PCA analysis was carried out. A biplot graphic representation of principal component scores PC1 and PC2 indicated that some of the pediatric patients were separated from the mild adult patients (AMD), and the severe adult patients (ASD) (Fig. 1B). Although some overlapping between groups was seen for a few individuals from each group, PC1 separated most of the ASD patients from the pediatric patients, while PC2 separated AMD patients from severe ones and pediatric patients (Fig. 1B). The absence of a perfect separation among groups of patients was not unexpected since some of the individuals from different groups may have similar cell frequency profiles. The PC scores from the three groups were plotted and the significance of the differences was evaluated by a KW test (Fig. 1C). Children had the highest mean score value for PC1 (2.765), followed by AMD (0.142) and ASD (−2.351), and these differences were highly significant (Fig. 1C). For PC2, AMD had a mean score of 2.079, higher ($p < 0.0001$) than children and ASD (mean scores of −1.585 and −1.180, respectively, $p > 0.999$). To further assess the immune responses of each group, we performed a hierarchical clustering analysis based on the original data. The results (Fig. 1E) showed three main clusters—highlighted in different shades of gray. The first cluster is mainly composed of adults with mild disease; the second cluster, mainly of children; and the third, mainly of adults with severe disease. In spite of the resulting clusters not being composed uniquely by individuals of the same group of patients, mild cases represent 72% of the cases in the first cluster whereas children slightly predominate in the second one, and ASD in the third one. Thus, the hierarchical clustering analysis supported the findings from PCA, showing that the immune responses of most of the children differ from adult individuals with severe COVID-19. We also examined principal components PC3, PC4 and PC5 scores for additional information about the differences in the profiles of child and adult patients. PC3 separated children from ASD, as already seen using PC1, whereas PC4 and PC5 did not separate the groups at all (Supplementary Fig. 2A). Principal components are calculated based on correlations among variables, and to interpret the meaning of each PC we analyzed the positive or negative contributions (loadings) of each variable in each PC. The variables with the main positive and negative contributions (loadings) are identified in Fig. 1D, for PC1 and PC2. Respective loadings values are listed in Supplementary Table 1. PC1 had positive inputs mainly by IgM+ memory B cells, naïve B cells, and cDC1 DR expression; and main negative contributions by proliferating B cells; plasmablasts; and CX3CR1+ expression in dendritic cells (DC). That indicated that the group with the highest mean scores for PC1 (children) would be characterized by a profile of predominantly naïve or low-affinity memory B cells,

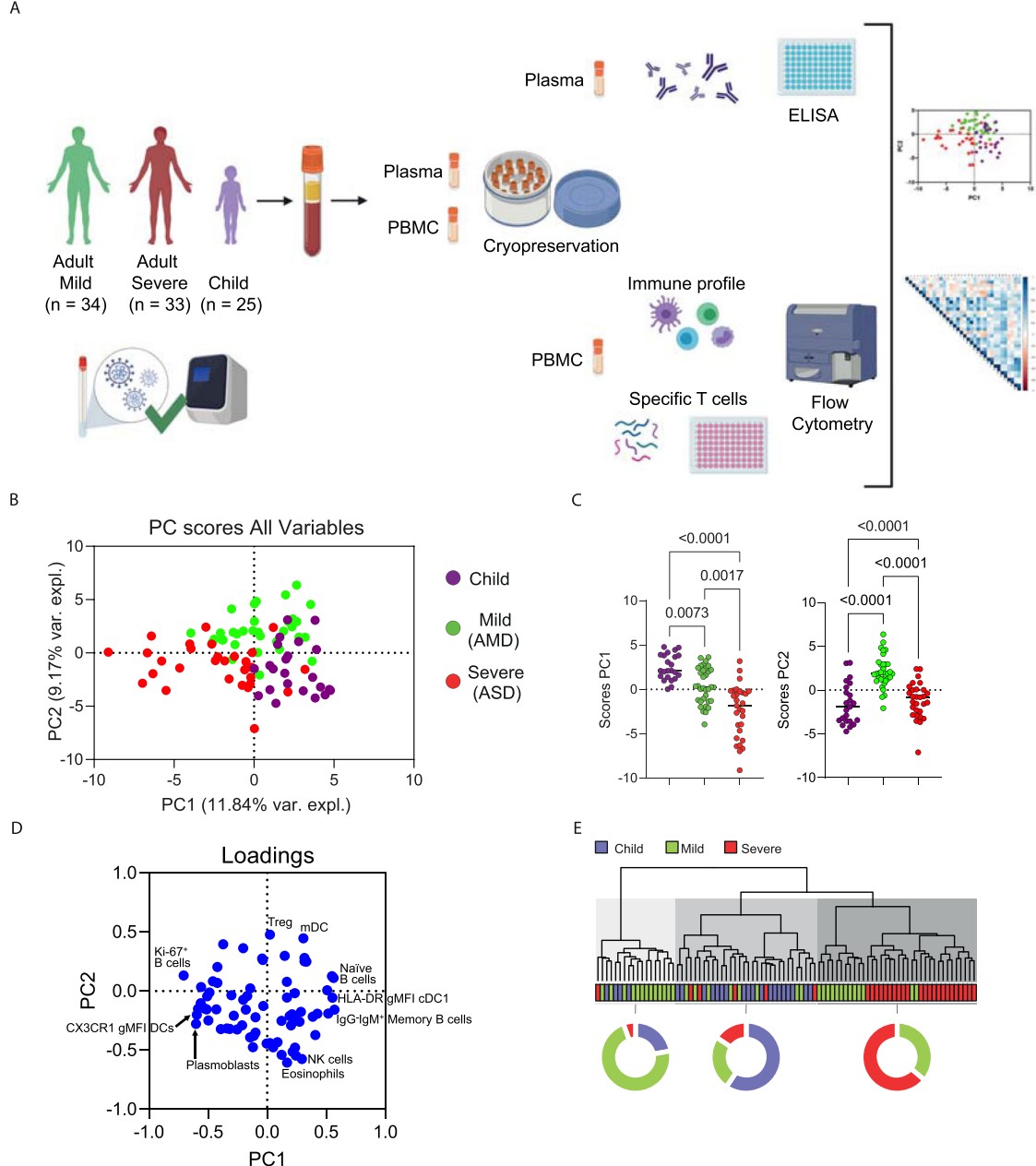

**Fig. 1 Experimental approach and differential immune profile of children, mild and adult patients by principal component analysis. A** Graphical representation of the study design; **B**–**E** Principal component analysis of the clusters of pediatric (purple) and adult patients with mild (green) and severe (red) disease; each dot represents a patient, color coded. **B** Distribution of clusters by PC1 and PC2; **C** comparison of scores for each PC by analysis of variance (Kruskal–Wallis) with *P* values indicated over brackets, the horizontal bar representing the median. **D** Contribution of variables (loadings) to PC1xPC2. Each blue dot is a variable. **E** Hierarchical clustering analysis of immune variables. Each shade of gray is a cluster with a representative distribution between groups. Graphical study design was created with biorender.com (accessed on 20 October 2021). Source data are provided as a Source Data file.

not activated/differentiated; and their dendritic cells would be high in DR, but low in CX3CR1. The opposite would be true for individuals with the lowest mean scores (ASD), while AMD would be characterized by an intermediate profile for these variables. The main positive influences for PC2 were T regs, myeloid dendritic cells (mDCs), and TEMRA cells, indicating mild adult patients present significantly higher frequencies of these cell subpopulations. The main negative influences for PC2 were eosinophils, natural killer (NK) cells, and granulocytes, and these should be the lowest in AMD. To verify these PCA interpretations, we compared them to the analysis of variance

(KW) results performed among the three groups of patients regarding the three most relevant, positive and negative (Fig. 2) influencers, variables for PC1 and PC2. Comparisons of the three groups for their variances regarding each variable agreed with the differences among them detected by PCA. What emerged from these two combined analyses was that AMD, ASD, and children differed based mostly on the state of activation of B and T lymphocytes, and in targeting innate inflammatory responses to inflamed tissues.

The percent of the total variability explained by these first principal components was low; the first two PCs together

**Table 1 Clinical characteristics of all patients in this study.**

| Characteristics | Mild (*n* = 34) | Severe (*n* = 33) | Children (*n* = 24) | *P*-value |
|---|---|---|---|---|
| Age (y), median (IQR) | 37.8 (27.0–44.6) | 60.8 (38.8–75.9) | 7.4 (2.5–13.8) | **2.20e-14**[a] |
| Female sex, *n* (%) | 22 (64.7) | 17 (51.5) | 10 (41.7) | 0.21[b] |
| Active or passive smoking, *n* (%) | 2 (5.9) | 6 (18.2) | 3 (12.5) | 0.08[b] |
| **Racial or ethnic group** | | | | |
| Caucasian, *n* (%) | 28 (82.4) | 20 (60.6) | 17 (70.8) | 0.22[b] |
| Non-caucasian, *n* (%) | 4 (11.8) | 1 (3.0) | 5 (20.8) | |
| **Days from symptom onset to sample collection** | | | | |
| Days, median (IQR) | 18.0 (16.0–20.5) | 10.0 (7.5–14.0) | 15.0 (7.8–17.0) | **1.00e-04**[a] |
| **Symptoms** | | | | |
| Headache, *n* (%) | 32 (94.1) | 23 (69.7) | 13 (54.2) | **0.02**[b] |
| Myalgia, *n* (%) | 30 (88.2) | 20 (60.6) | 8 (33.3) | **1.83e-03**[b] |
| Malaise, *n* (%) | 28 (82.4) | 31 (93.9) | 14 (58.3) | **1.26e-03**[b] |
| Coryza, *n* (%) | 26 (76.5) | 18 (54.5) | 15 (62.5) | 0.21[b] |
| Cough, *n* (%) | 25 (73.5) | 30 (90.9) | 16 (66.7) | 0.07[b] |
| Fever, *n* (%) | 23 (67.6) | 26 (78.8) | 19 (79.2) | 0.51[b] |
| Chills, *n* (%) | 21 (61.8) | 20 (60.6) | 9 (37.5) | 0.12[b] |
| Dyspnea, *n* (%) | 20 (58.8) | 22 (66.7) | 4 (16.7) | **4.57e-04**[b] |
| Dysgeusia, *n* (%) | 20 (58.8) | 12 (36.4) | 6 (25.0) | 0.19[b] |
| Sore throat, *n* (%) | 19 (55.9) | 12 (36.4) | 10 (41.7) | 0.32[b] |
| Appetite loss, *n* (%) | 19 (55.9) | 21 (63.6) | 12 (50.0) | 0.56[b] |
| Anosmia, *n* (%) | 19 (55.9) | 11 (33.3) | 6 (25.0) | 0.21[b] |
| Stuffy nose, *n* (%) | 17 (50.0) | 11 (33.3) | 13 (54.2) | 0.33[b] |
| Conjuctivitis, *n* (%) | 16 (47.1) | 10 (30.3) | 7 (29.2) | 0.28[b] |
| Nausea, *n* (%) | 14 (41.2) | 12 (36.4) | 6 (25.0) | 0.64[b] |
| Sputum production, *n* (%) | 12 (35.3) | 10 (30.3) | 6 (25.0) | 0.70[b] |
| Diarrhea, *n* (%) | 12 (35.3) | 16 (48.5) | 7 (29.2) | 0.25[b] |
| Vomiting, *n* (%) | 2 (5.9) | 4 (12.1) | 4 (16.7) | 0.42[b] |
| Skin rash, *n* (%) | 1 (2.9) | 1 (3.0) | 1 (4.2) | 0.97[b] |
| **Underlying medical conditions** | | | | |
| Obesity, *n* (%) | 10 (29.4) | 13 (39.4) | 0 (0.0) | **0.01**[b] |
| Hypertension, *n* (%) | 6 (17.6) | 15 (45.5) | 0 (0.0) | 1.00[b] |
| Asthma, *n* (%) | 1 (2.9) | 2 (6.1) | 5 (20.8) | **3.77e-05**[b] |
| Diabetes mellitus, type 1 and 2, *n* (%) | 1 (2.9) | 11 (33.3) | 0 (0.0) | **0.04**[b] |
| Cancer, *n* (%) | 1 (2.9) | 2 (6.1) | 0 (0.0) | 0.73[b] |
| Tuberculosis, *n* (%) | 0 (0.0) | 1 (3.0) | 0 (0.0) | 0.73[b] |
| Stroke/CVA, *n* (%) | 0 (0.0) | 4 (12.1) | 0 (0.0) | 0.26[b] |
| COPD, *n* (%) | 0 (0.0) | 4 (12.1) | 1 (4.2) | 0.43[b] |
| Heart failure, *n* (%) | 0 (0.0) | 2 (6.1) | 0 (0.0) | 0.53[b] |
| Congenital heart disease, *n* (%) | 0 (0.0) | 1 (3.0) | 0 (0.0) | 0.73[b] |
| **Ct value (median, IQR)** | | | | |
| ORF1ab | 19.3 (16.7–22.6) | 24.3 (19.9–29.7) | 19.0 (15.4–27.5) | 0.05[a] |
| S | 19.8 (16.7–23.0) | 25.3 (21.9–27.7) | 19.0 (13.2–28.3) | **0.02**[a] |
| N | 18.7 (16.1–23.5) | 24.0 (20.6–28.6) | 19.0 (14.1–29.2) | **0.02**[a] |
| **Oxigen use** | | | | |
| Oxigen use during hospitalization, *n* (%) | 0 (0.0) | 22 (66.7) | 0 (0.0) | **8.44e-12**[b] |

IQR = interquartile range.
[a]Kruskal–Wallis test.
[b]Pearson's Chi-squared test; Significance *P*-value are bold.

explained only 21% of the variance (Supplementary Table 1). This indicated that these 77 variables were not highly correlated. A correlation analysis using Spearman's coefficient confirmed this observation (Supplementary Fig. 2), revealing a general pattern of moderate to weak correlations, but also identifying clusters of variables that were more correlated than others. These clusters represented six types of immune "signatures": proliferating/ activated T cells; DCs; granulocytes + monocytes; NK cells; B cells; and memory T cells. Follicular helper T cells (Tfh) related variables were weakly correlated and were not considered as a cluster. To further identify differences and/or similarities of the general immune responses among the three groups of COVID-19 patients we performed six separated PCAs for the identified clusters of variables. In this analysis, the first two PCs for each cluster were now explained a larger portion of the total variance (44.19, 63.17, 50.37, 67.35, 49.56, and 42.49%, respectively -

Supplementary Table 1) among the three types of patients, and their distributions in children, AMD and ASD were analyzed. The results are shown in Fig. 3A–D and Supplementary Fig. 3A–C, with the respective graphic representations for scores and loadings. We started by analyzing PCs formed by the innate cells' signatures. Principal components for the clusters of Granulocytes + Monocytes (Fig. 3A) and NK cells (Fig. 3B), although derived from expressive correlations among their respective variables, did not separate the three groups of patients, indicating that the individuals were not significantly different for the variables that composed these PCs, even though they were highly correlated (Supplementary Fig. 2B, C). That was intriguing because those variables had, as stated above, important contributions for the PC2 of all variables (Fig. 1), but it also indicated that this contribution helped separate the groups mostly based on their correlations with other variables in the group of all variables,

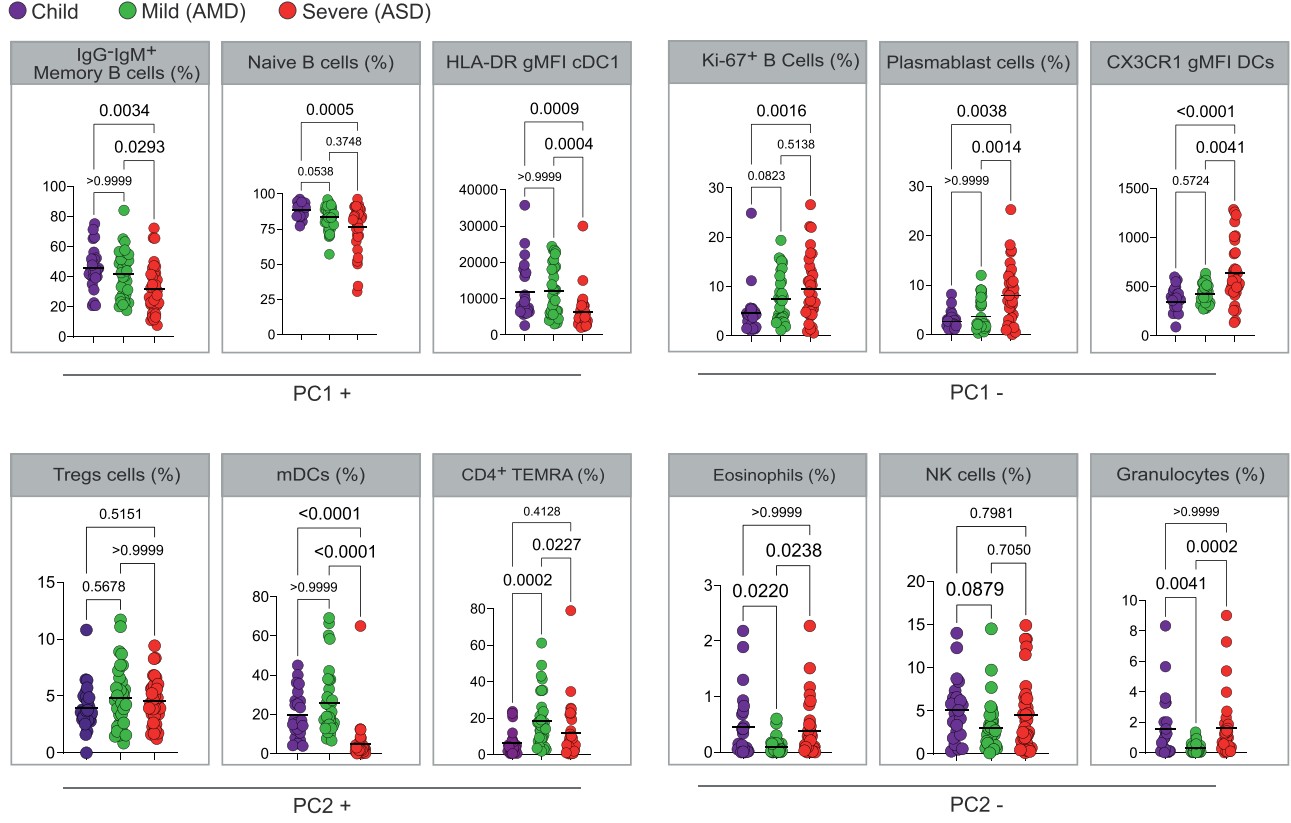

**Fig. 2 Analysis of variance of the main variables contributing to principal components. A, B** Kruskal–Wallis tests comparing values of each of the three immune variables that presented the highest influences—either positive (**A**) or negative (**B**) for PC1, PC2 and PC3. Each dot represents a patient, color coded: children—purple, adult with mild disease—green, and adult with severe disease—red. *P* values are indicated over brackets, the horizontal bar representing the mean. Source data are provided as a Source Data file.

and not on the differences among groups for those variables alone.

Findings in PCA for the DCs signature (Fig. 3C) indicated that children were significantly separated from ASD, but not from AMD, with lower scores for PC1 (mostly CX3CR1 expression in DCs) and higher scores for frequencies of DCs (Fig. 3C). HLA-DR expression in DCs subpopulations constituted negative contributions for PC1. This indicated that children, as well as AMD, would have more DCs than ASD, high in HLA-DR and low in CX3CR1. That was confirmed by the KW analysis of the three groups (Fig. 3C). Spearman correlation analysis (Fig. 3D) evidenced that CX3CR1 expression was negatively correlated with HLA-DR expression. These results suggested that high DCs frequencies in blood, with low CX3CR1, but high HLA-DR expression could be involved, or at least serve as markers, for mild disease. Conversely, low DCs frequencies, with high CX3CR1 expression, could be associated with more severe disease.

PCA for the clusters involving adaptive cells variables was performed next and are shown in Supplementary Fig. 3. B cells (Supplementary Fig. 3A, Supplementary Table 1) and T cell activation/proliferation (Supplementary Fig. 3B, Supplementary Table 1) PCA corroborated a general pattern of response in children, either separating from the other groups (only sometimes grouping with AMD, apart from ASD). KW analysis of mean scores for each PC showed that in some cases children presented some significant differences from AMD. For example, PC1 of B cells recapitulated findings from the first analysis, mainly positively influenced for IgM+ B cells and naïve cells, and showed children with significantly higher scores, compared to mild and ASD ($p < 0.01$). Children were grouped with AMD ($p = 0.1599$), and apart from ASD ($p < 0.0001$), regarding PC1 of

T cell activation/proliferation (Supplementary Fig. 3B). The ASD had the highest scores for this PC, highly influenced by activated and proliferating CD4+ and CD8+ T cells, suggesting that adults with severe disease were characterized by higher frequencies of activated, Ki67+ T cells, and the opposite would be observed for children and AMD—though this was not always corroborated by the KW analysis. Finally, PCA for T cell memory clustered variables showed a trend to separate the three groups (Supplementary Fig. 3C). PC1 scores, strongly positively influenced by naïve T cells, but negatively influenced by effector memory T cells (TEM), significantly separated children from AMD ($p = 0.0005$), and these somewhat separated from ASD ($p = 0.0410$), indicating children and AMD would have lower frequencies of TEM and higher frequencies of naive T cells compared to ASD. Terminally differentiated memory CD4+ and CD8+ T cells (TEMRA) were strong positive influences for PC2, while expression of CD69 and CD137 in TEM cells negatively influenced PC2. Children and AMD, with high scores for PC2, did not differ from each other, suggesting they would both be characterized by higher frequencies of TEMRA (especially CD4+ TEMRA) cells than ASD, which in turn would have higher frequencies of activated, CD69+, CD137+, TEM cells. Confirmations of the interpretations of these PCs were again sought in the KW analysis for individual variables next to each PCA result (Supplementary Fig. 3C), and in Supplementary Fig. 4—which compiles all the remaining variables KW analyses results. This led us to note that for TEMRA, children differed from AMD and not from ASD. CD45RA, a marker upregulated both in naïve and TEMRA cells, has been reported to show varied expression during the generation of memory pools of chronic infections[19] as well as in response to vaccination[20]. CD4+ TEMRA cells associated with

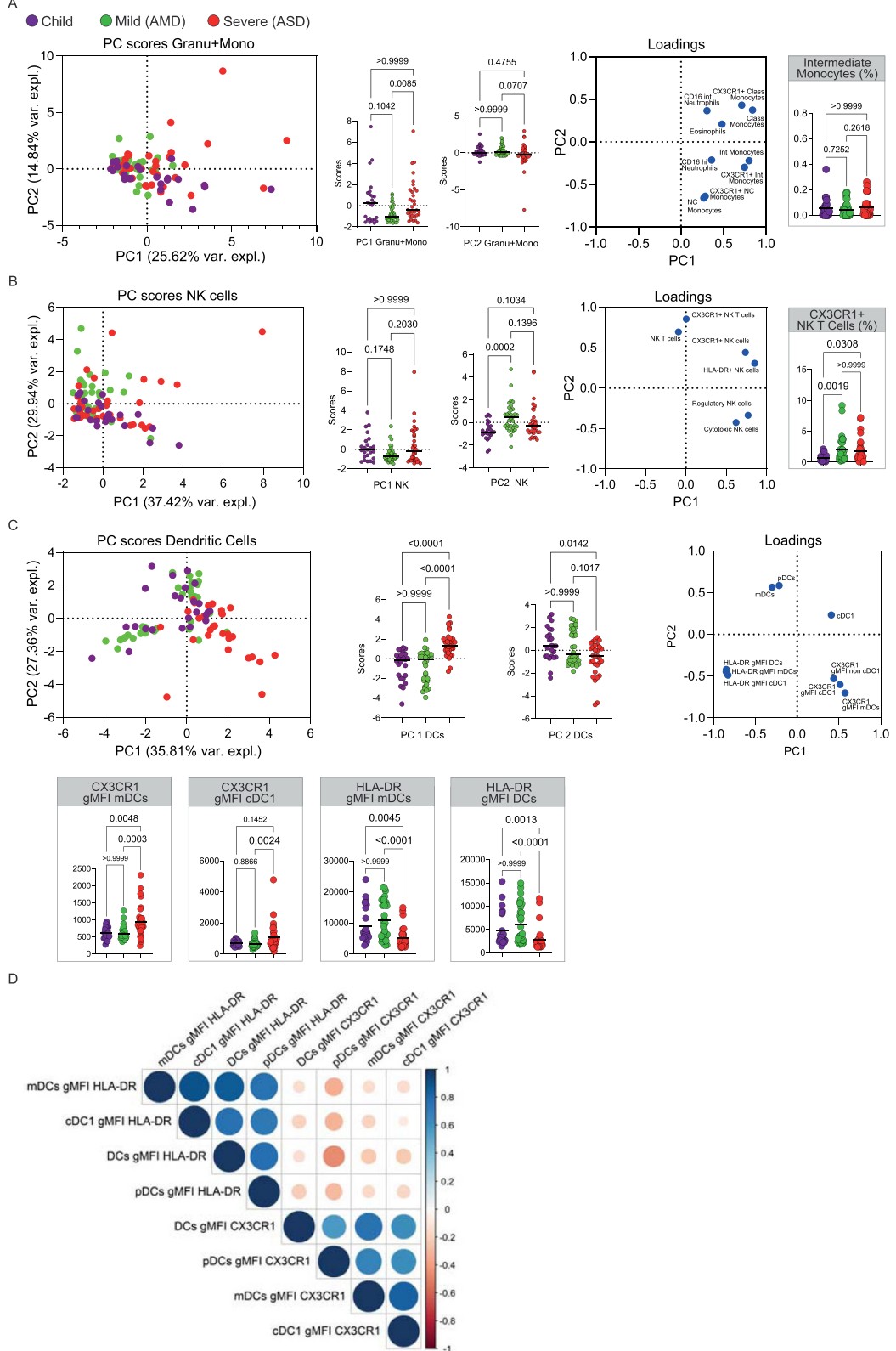

**Fig. 3 Principal Component Analysis of innate cells immune signatures. A–G** Principal component analysis of the clusters of patients (each dot representing a patient color coded), according to the immune signatures (**A** Granu+Mono, Granulocytes and Monocytes; **B** NK cells; **C** Dendritic Cells; **D** Spearman correlation analysis of HLA-DR and CX3CR1 expression in DCS. For each signature, are displayed the PCA plot of PC1xPC2, the differences in scores of individuals for each PC; the loadings of the main variables contributing to each PC and Kruskal–Wallis tests comparisons of the scores of the major contributing variables values for each group of patients. *P* values are indicated over brackets, the horizontal bar representing the median for scores; and the mean for percentages and gMFI. Source data are provided as a Source Data file.

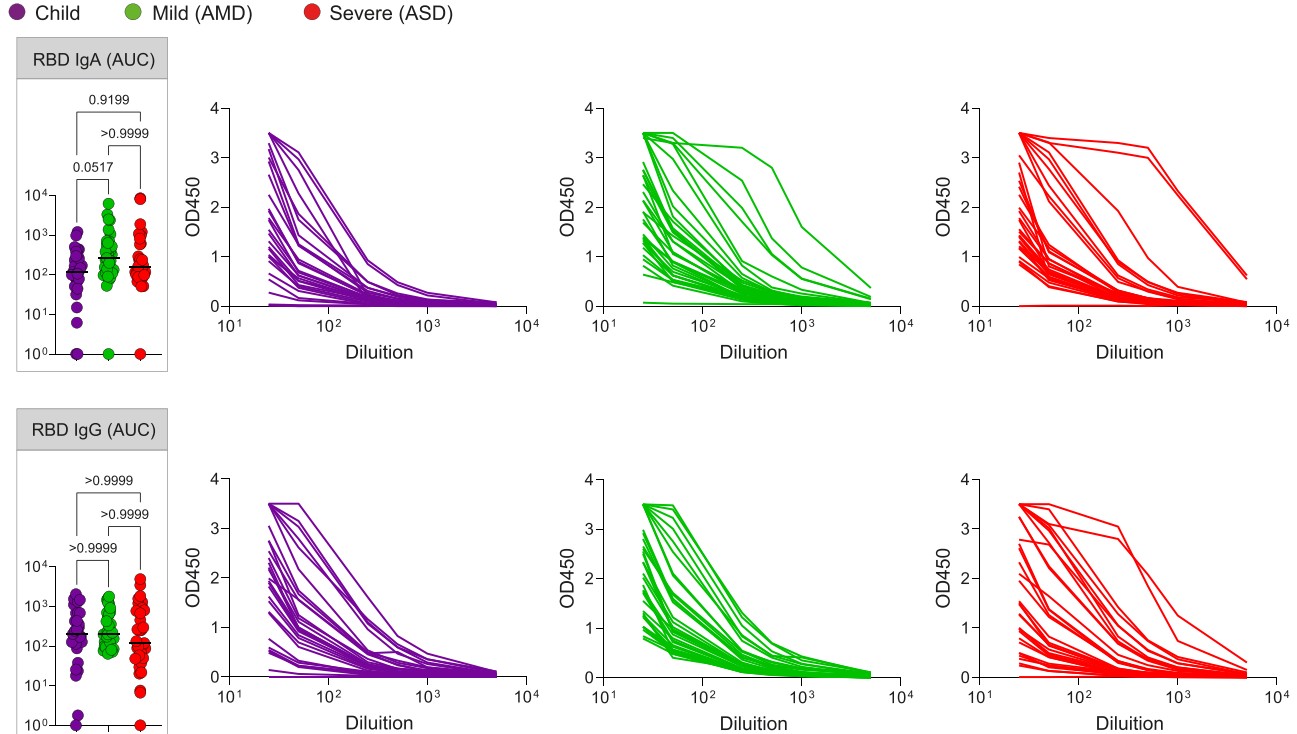

**Fig. 4 Antibody responses.** SARS-CoV-2 spike RBD IgA and IgG antibody titers determined by ELISA using serial dilutions of plasma. Individual titration curves for individuals (represented by a line, color coded) and analysis of variance (Kruskal-Wallis) of the values calculated as the area under the curve (AUC) for IgA (**A**) and IgG (**B**) are displayed. P values are displayed over brackets, the horizontal bar representing the median. Source data are provided as a Source Data file.

protection in dengue[21]. Thus, the PCA could indicate possible differences for children and AMD, compared to ASD, in pathways for the generation of memory. In the KW comparisons, CD4+ TEMRA cells were lower in children than in AMD, and only AMD differed from ASD significantly. For central memory T cells (TCM), there were no differences among groups (Supplementary Fig. 4A). Altogether, the patterns revealed by PCA indicated that children have higher frequencies of non-specific antigen inexperienced B and T cells and DCs, with high HLA-DR and low CX3CR1 expression. In some cases, children and AMD shared not only a mild presentation of the disease but also a similar immune profile. The immune profile of ASD was characterized by higher frequencies and markers of T and B cell activation and proliferation, TEM cells, and lower DCs with high expression of CX3CR1. Finally, we investigated the production of inflammatory plasma cytokines (Supplementary Fig. 4B). Mostly, only subjects with severe disease showed, at least on average, significant higher levels of detectable cytokines in blood, except for IL-4, which was increased in children compared to AMD.

**SARS-CoV-2 specific T cells and antibodies responses in children are comparable to the ones of adult patients.** The characteristics of the non-specific immune profile of children led us to ask if they had effectively formed SARS-CoV-2 specific responses upon infection. Seroconversion after infection with SARS-CoV-2 patients with all forms of the disease has been described by several studies—reviewed in[22]. Antibodies to the S protein, and more specifically to the RBD of this protein, are clinically considered a hallmark of infection, and frequently proposed as a correlate of protection. We thus compared children, AMD, and ASD for their RBD specific—IgA and IgG titers. On average, children presented levels of both IgG and IgA comparable to the adult patients

(Fig. 4). In our cohort, although some individuals from the ASD group presented higher levels of antibodies, differences among the groups were non-significant. Our results revealed that, even though children presented a generally naïve, non-activated, immune profile, they had efficiently generated SARS-CoV-2 specific antibody responses, in levels that did not differ from the ones in AMD ($p < 0.103$ for IgA; $p > 0.999$ for IgG) or ASD ($p < 0.916$ for IgA; $p > 0.999$ for IgG) COVID-19 patients.

We next asked if children had generated specific effector T cell responses to SARS-CoV-2. There are four structural proteins in SARS-CoV-2: the spike glycoprotein (S), the envelope (E) protein, the membrane (M) protein, and the nucleocapsid (N) protein. Specific effector T cell responses have been described in adult COVID-19 patients, both with mild and severe disease[23,24], however fewer studies have focused on specific immune responses in pediatric patients infected with SARS-CoV-2. We measured the frequencies of CD4+ T and CD8+ T cells expressing tumor necrosis factor (TNF), gamma-interferon (IFNγ), or interleukin-17 (IL-17) in response to stimulation by peptide pools of the S, N, and M proteins of the virus (Fig. 5). Figure 5A shows representative flow cytometry plots of cytokine-producing CD4+ or CD8+ T cells upon stimulation with SARS-CoV-2 peptide pools. Negative (DMSO) and positive (PMA + ionomycin) control representative plots can be seen in Supplementary Fig. 5. Children presented detectable CD4+ and CD8+ T cell responses upon stimulation with all three peptide pools (Fig. 5B). When we compared types of responses in each group for the different peptide pools, children showed a significantly higher TNF+CD8+ T response for the M ($p < 0.005$) and for the N ($p < 0.0409$) peptide pools than for the S pool (5B, upper right panel). This was not seen in adults, and although there was a trend for lower CD4+TNF+ responses in children, it was not significant. Supplementary Fig. 6 shows the responses compared

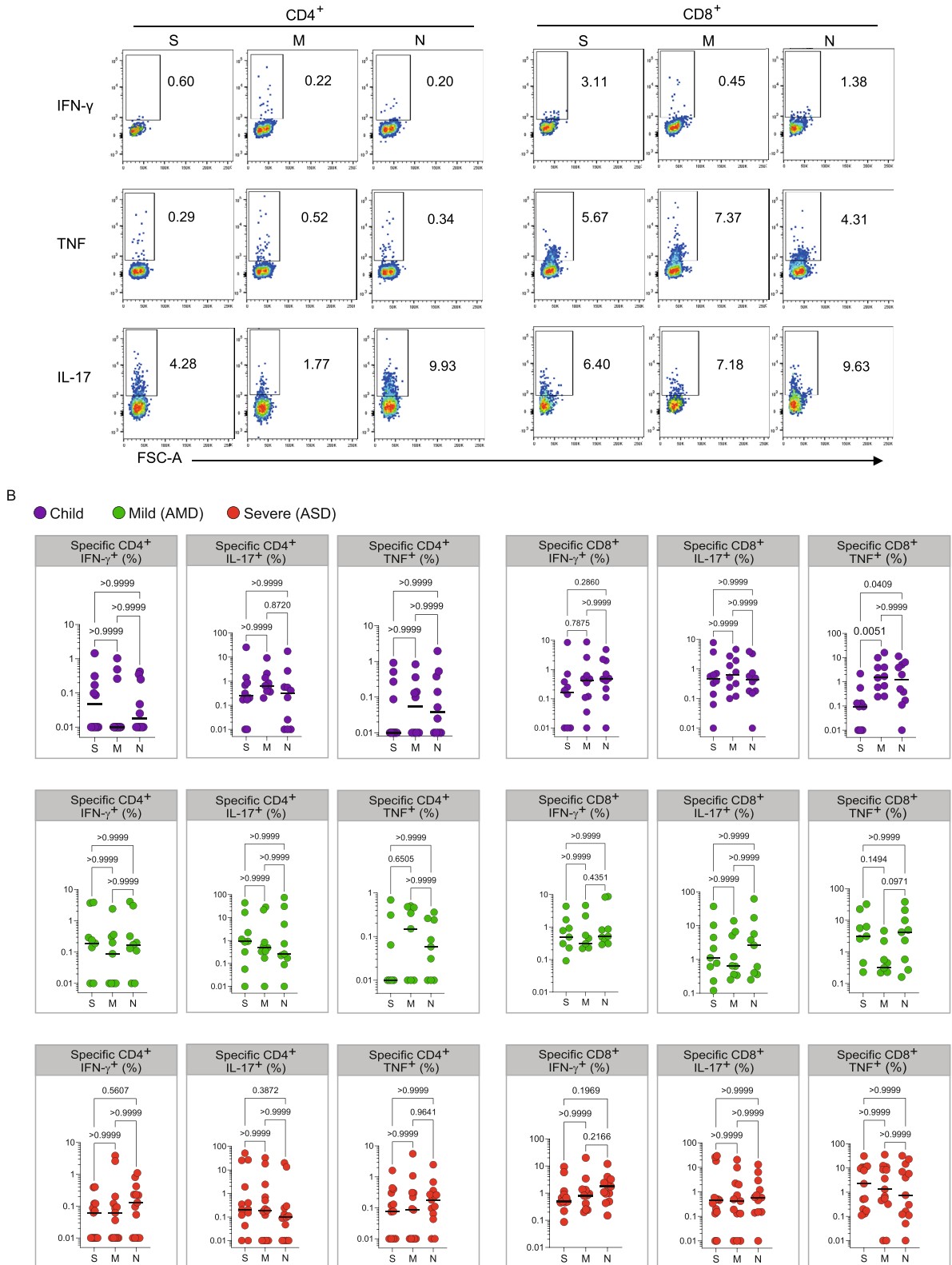

**Fig. 5 Specific T cell responses. A** Gating strategies and typical plots of CD4$^+$ and CD8$^+$ T cells stimulated with peptide pools from structural proteins spike (S), membrane (M) and nucleocapsid (N), and analyzed by flow cytometry for cytokine production. **B** Comparisons of effector T cells in each group— percentages of CD4$^+$ or CD8$^+$ cells, producing IFNγ, TNF or IL-17 in response to stimulation by each peptide pool. Each dot represents a patient, color coded: purple for children; green for adults with mild disease and red for adults with severe disease. All analyses are Kruskal–Wallis tests, and the $p$ values are indicated over brackets ($n$ children = 10, $n$ AMD = 9, $n$ ASD = 13). Significant differences are indicated by $p$ values in a higher font, the horizontal bar representing the median. Source data are provided as a Source Data file.

among the groups. About 30% of individuals—of all groups—did not show responding CD4[+] T cells to the peptide pools; a higher frequency of individuals did not respond to the S pool compared to the M and N pools. In the ones that responded, CD4[+] IL-17[+] T cell responses were higher for all three peptide pools (about 1 log higher than IFNγ and TNF CD4[+] T cell responses). CD8[+] T cell responses were, in general, more robust, although for S and M peptide pools there were still some individuals, though fewer, that did not respond to stimulation. The absence of response, in our sample, did not correlate with the early time of collection, as reported by[19]. TNF[+]CD8[+] T cell responses were about 1 log higher than what was detected for CD4[+] T cells, for all peptide pools (Supplementary Fig. 6). The IL-17[+]CD4[+] T cell responses to stimulation by all three pools were higher than the TNF[+]CD4[+] and IFNγ[+]CD4[+] for all three groups. The differential TNF[+] cytotoxic response to M and N peptides seen in children led us to investigate levels of anti-N antibodies. Children made strong anti-N IgG levels, not different from AMD and ASD (Fig. 6A). Anti- RBD IgA, but not IgG levels, correlated positively with CD4[+] IFNγ[+] responses (Fig. 6B). Interestingly, the TNF[+] cytotoxic responses to M and N peptide pools were inversely correlated with levels of anti-RBD and anti-N antibodies (Fig. 6B). Anti-N antibody levels correlated positively with anti-N CD4[+] IFNγ[+] responses (Fig. 6B). Taken together, these results indicate that children do generate specific humoral and effector cell responses upon infection with SARS-CoV-2, with a differential, higher cytotoxic response against proteins M and N, not associated with antibody responses to the spike protein.

## Discussion

It is clear from our study as well as from others[25] that children do get infected by SARS-CoV-2, and thus possibly contribute to the community-based spread of the virus, contrary to what is suggested by studies on the low nasal ACE2 expression in children[26]. The lower rates of infection in children can be biased by lower testing, as pointed out by[27], and should be more carefully studied, given its importance for planning school openings. Our findings on the more naïve, non-specific lymphocyte profile presented by in children, if taken isolated from the other results in this study, may indicate that a naïve immune system does better than an old one, as has been suggested[9] and that children would be better equipped to mount fast and efficient immune responses to rapidly clear the virus. However, a prediction from this hypothesis is that children would be more likely to present mild forms of all viral diseases, and this is not the case. While milder manifestations are observed in Middle East respiratory syndrome (MERS), SARS, and varicella, the opposite is observed for infection with poliovirus, and respiratory viruses, especially influenza and respiratory syncytial virus (RSV) - reviewed in[6].

Our data indicate that not only the adaptive but also their innate immune system has characteristics that enable children to mount an immune response that controls the infection. The differences observed for dendritic cells might offer an important clue. DCs play crucial roles in initiating and shaping the adaptive response, and subpopulations of DCs, especially pDC, are determinant for the generation of efficient antiviral responses, being one of the main sources of type I interferon[28]. A previous study in COVID-19 adult patients indicated decreased activation and numbers of DCs[29]. In our study, children consistently show higher frequencies of DCs, including pDC, compared to adults. High HLA-DR expression is characteristic of mature DCs; as they become activated by engagement of pattern recognizing receptors, HLA-DR first increases, and then decreases as DCs migrate to draining lymph nodes[28,30]. Low HLA-DR expression in children is associated with immune suppression[4] and acute inflammatory conditions[31].

The inverse correlation of HLA-DR with CX3CR1 in DCs is intriguing. CX3CR1, also known as the fractalkine receptor, is considered a homing marker for inflamed tissue and plays a role in pathology in Japanese virus-induced encephalitis[32] and peritoneal vasculitis in a sepsis model[33]. The high levels of HLA-DR in children's DCs indicate that their cells are not poorly activated, but mature and able to generate efficient immune responses. Heinonen et al. integrated blood transcriptomics and cell profiling with clinical data in a cohort of 190 children with RSV and found low HLA-DR in monocytes to be associated with severe disease[34]. The low expression of CX3CR1 in our samples suggests that children's DCs are not targeted to inflamed sites, as they seem to be in ASD. Our results indicate that expression of CX3CR1 in circulating DCs could associate, or serve as a marker for, pathological mechanisms in severe COVID-19, suggesting that inflammation to specific sites such as the lung may be affected by age, and that low CX3CR1 expression in DCs might favor the generation of antiviral responses. Loske et al.[35], in a single-cell transcriptomics profile of nasal samples, suggested a pre-activated state of the antiviral innate immune cells in children - DCs expressing higher levels of MDA5 and RIG I, two pattern recognition receptors relevant for antiviral responses. Also, children showed a KLRG1-expressing population of memory CD8[+] T cells. Both data concur with our findings, which support a role for potentially responsive DC and effector cytotoxic T cell populations in the differential, possibly protective, immune response in children that is associated with milder disease.

Pediatric patients in our sample presented SARS-CoV-2 specific antibodies and T cell responses in levels comparable to adult patients. The higher TNF[+]CD8[+] responses for the M and N proteins could be associated with a protective response in children. The M protein is the most abundant structural protein on the surface of the virus[36], potentially constituting an important target of immune responses. A study by Thieme et al. found anti-M CD4[+] T cells as the highest T cell response in critical COVID-19 patients[37]. In that study, CD4[+] T, rather than CD8[+] T cell immunity to SARS-CoV-2 proteins dominated the response in severe and critical patients, indicating that a robust CD4[+] T cell response to these antigens did not correlate with protection. The nucleocapsid (N) protein is structural, abundantly produced upon infection, and highly conserved among beta coronaviruses[38]. It was a major target for early B cell responses in the SARS epidemic of 2003[39]. Recent studies[18,40] found strong anti-N T cell responses in SARS-CoV-2 patients, and in individuals who recovered from SARS, supporting a relevant role for structural N protein as an immune target in SARS-CoV-2 infection. We found robust antibody responses in children both against the S RBD and N protein, while Weisberg et al.[9] found antibodies to the S, but not the N protein; and Cohen et al.[41] found lower responses in children in general. These differences, as well as a lower response to the S protein in general in our sample, may reflect differences in HLA between American and Brazilian populations, or even a difference in immunization history, given a tradition in vaccination programs for children in Brazil. The correlations of antibody responses with CD4[+] T cells are somewhat expected, given the help needed for antibody production, and indicate that these responses are somewhat coordinated, but also that not all antibodies produced are linked to TNF or IFNγ help. The inverse correlation between specific CD8[+] T cell responses and antibodies may indicate a relevant role for cytotoxic immunity against SARS-CoV-2, beyond antibody production.

A hypothesis frequently raised to explain milder disease in children with COVID-19 is that the presence of neutralizing antibodies to such viruses could cross-protect them upon infection with SARS-CoV-2. However, a recent study in adults found no evidence of cross-protection associated with levels of these antibodies[42]. Alternatively, protection could be conferred

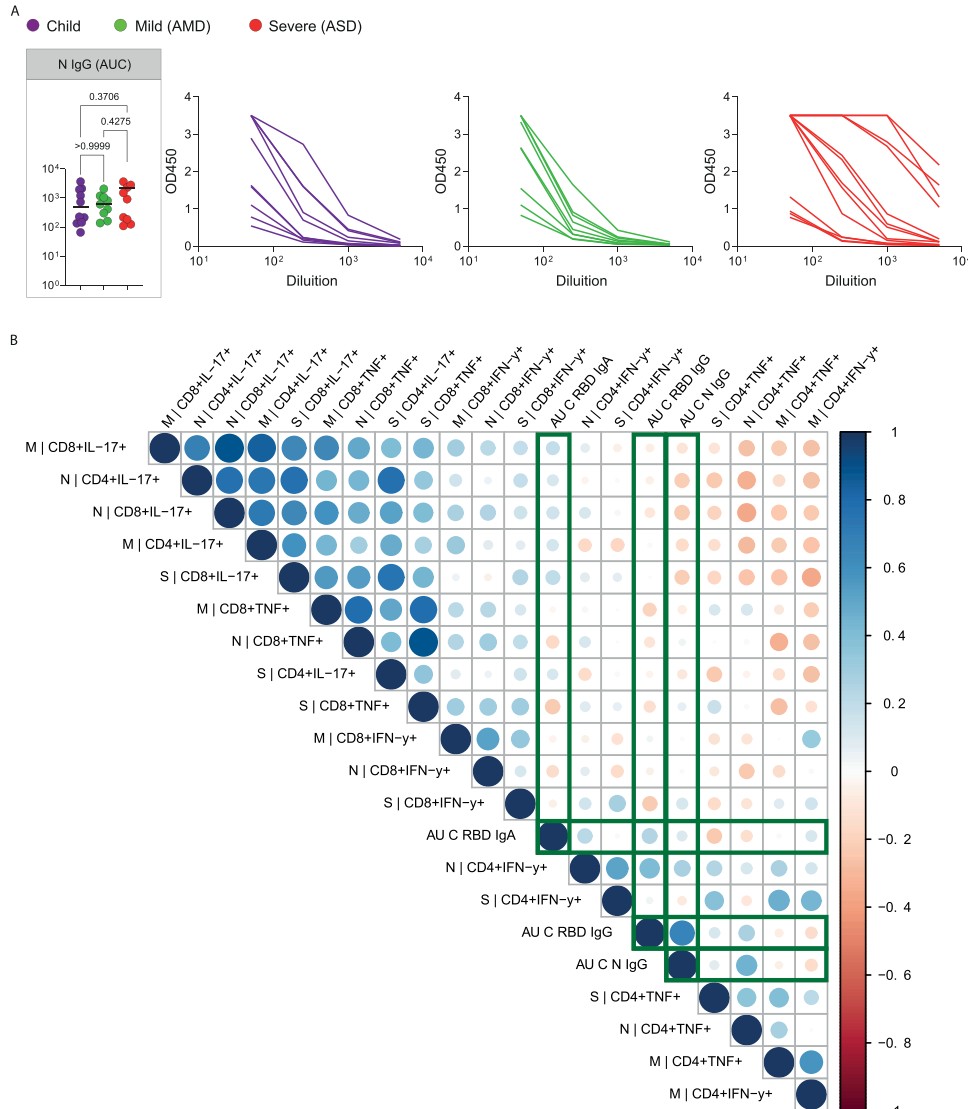

**Fig. 6 Anti-N IgG response and correlation of specific responses. A** SARS-CoV-2 anti-N IgG antibody titers determined by ELISA using serial dilutions of plasma. Individual titration curves for individuals (represented by a line, color coded) and analysis of variance (Kruskal–Wallis) of the values calculated as the area under the curve (AUC) for IgG (**A**) are displayed. *P* values are displayed over brackets, the horizontal bar representing the median. **B** Matrix representing a Spearman correlation analysis of specific effector T cells responses (in percentages of CD4[+] and CD8[+] cells expressing citokines in response to peptide pools) and the antibody response to the RBD of the spike protein and to the N protein (represented as values for the AUC). Correlations between the specific effector CD8[+] T and CD4[+] T cells frequencies, and the antibody (AUC—area under the curve) values for the RBD and N protein are highlighted. Source data are provided as a Source Data file. *n* children = 10, *n* AMD = 9, *n* ASD = 13.

not by cross-reactive antibodies, but rather by pre-existing N-protein-specific T cells. Most studies—and most vaccines—have so far focused on protection against SARS-CoV-2 infection by antibodies to the spike protein. Both screen studies by Ferretti et al.[40] and Ng et al.[10] indicated that T cell immunity to SARS-CoV-2 infected individuals includes many targets outside the spike protein and that they are not conserved among coronaviruses that cause the common cold. These findings agree with the ones of the Le Bert study[24]. Our results support that the role of T cell responses to the N protein must be further investigated, with a more detailed T cell epitope mapping. Such work might reveal additional correlates of protection, and/or epitopes to add in the next generation of COVID-19 vaccines. A recent report[43] indicated that T cell immunity was not markedly affected, so far, by the emergence of new variants, supporting the identification of T cell epitopes to be added to the next vaccines.

The main limitation of this study is that it is mostly an exploratory, descriptive one, and compares individuals in different age groups. A second important limitation is the small sample size. We believe this was a valid approach given the magnitude of what is still unknown regarding pediatric immunity in COVID-19. Biomarkers in peripheral blood are only useful when highly correlated with outcomes. Yet, most studies that seek to understand how immune responses can correlate with protection compare adults with mild and severe disease, and it is known that these are in different age groups. At present, the best age-matched controls for SARS-CoV-2 infected patients are still unknown. Certainly, the absence of a pre-pandemic healthy children control group is a limitation of this study and all the other ones that focused on the general, non-specific immune profile of children with COVID-19. There is still much to be understood about immunological differences not only between pediatric and adult COVID-19 patients but also in other diseases. At the time the

project started, COVID-19 numbers in Brazil were still not high and the frequency of MISC patients or children with severe manifestations of the disease was still too low to include. The study on this cohort is still ongoing, with two more points of sample collection. We expect that further analysis of our data, as well as other studies, on immune profile data and specific responses, will bring relevant information on the generation of immune memory in pediatric COVID-19.

## Methods

**Ethics statement.** This study was approved by the Institutional Review Board (IRB 30749720.4.1001.5330) at Hospital Moinhos de Vento and Ethics Committee from Fundação Faculdade Federal de Ciências Médicas de Porto Alegre (CEP-UFCSPA) (CAAE 30749720.4.3001.5345). Informed consent was obtained from all participants or their legal guardians. The study was conducted according to good laboratory practices and following the Declaration of Helsinki.

**Patients.** A prospective cohort study was carried out at Hospital Moinhos de Vento and at Hospital Restinga e Extremo Sul, both in Porto Alegre, southern Brazil. A convenience sample of adults and children older than 2 months were enrolled from June to December 2020 at either the outpatient clinics (OPC), emergency rooms (ER, or hospitalized. Subjects were screened if presenting cough and/or axillary temperature ≥37.8 °C and/or sore throat. Both blood samples and respiratory samples collected through nasopharyngeal swabs were obtained at enrollment. Only patients with the clinical diagnosis of COVID-19 and SARS-CoV-2 infection confirmed by RT-PCR were included in the study. Clinical and demographic data were collected at inclusion, following a standardized protocol. Disease severity was classified according to the World Health Organization classification following a standardized protocol. Disease severity was classified accordingly after completing the follow-up questionnaire[16].

**SARS-CoV-2 RT-q-PCR.** A qualitative RT-PCR assay to SARS-CoV-2 was performed for all participants. Bilateral nasopharyngeal and oropharyngeal swabs were collected and placed in the same transport medium with saline solution and RNAlater®, RNA Stabilization Solution (Catalog number AM7021, Invitrogen™). MagMax™ Viral/Pathogenic Nucleic Acid Isolation Kit (Applied Biosystems) was used to extract viral RNA in the KingFisher Duo Prime System (ThermoFisher, USA) automated platform. The RT-PCR assay was performed in 10 μL total reaction, using Path™ 1-Step RT-qPCR Master Mix, CG (catalog number A15299, AppliedBiosystems) and TaqMan™ 2019-nCoV Assay Kit v1 (catalog number A47532, AppliedBiosystems) which comprises the SARS-CoV-2–specific targets (gene ORF1ab, gene S and gene N). As reaction control, we used 5 μL (200 copies/μL) of the TaqMan™ 2019-nCoV Control Kit v1 (catalog number A47533, AppliedBiosystems). QuantStudio 5 (ThermoFisher Scientific, USA) was used to perform the PCR.

**PBMC isolation and cryopreservation.** Blood was collected in EDTA tubes (Firstlab, PR, Brazil) and stored at room temperature before processing for PBMC isolation and plasma collection. Plasma was separated by centrifugation and cryopreserved. PBMCs were next isolated by density-gradient centrifugation using Ficoll–Paque™ PLUS (GE Healthcare®), and either studied directly or resuspended in fetal bovine serum (FBS) 5% DMSO and stored in liquid nitrogen until use.

**Flow cytometry.** Cells were thawed by diluting them in 5 mL pre-warmed complete RPMI1640 medium (Sigma-Aldrich - R8758) containing 5% FBS and spun at 1500 rpm for 5 min. Supernatants were carefully removed, and cells were resuspended in PBS. After, were stained with the BD Horizon™ Fixable Viability Stain 510 together with antibodies for surface markers, as follows: anti-CD3-APC-H7 (clone SK7), anti-CD24-APC-H7 (clone ML5), anti-HLA-DR-APC-H7 (clone G46-6), anti-CD4-PerCP-Cy5.5 (clone RPA-T4), anti-CD27-PerCP-Cy5.5 (clone M-T271), anti-CD11c-PerCP-Cy5.5 (clone B-ly6), anti-CD14-PerCP-Cy5.5 (clone M5E2), anti-CD8-FITC (clone HIT8a), anti-IgG-FITC (clone G18-145), Lineage 2-FITC (cat. 643397), anti-CD16-FITC (clone 3G8), anti-CXCR5 (CD185)-BB515 (clone RF8B2), anti-CD19-APC (clone HIB19), anti-CD127-Alexa 647 (clone HIL-7R-M21), anti-CX3CR1-Alexa647 (clone 2A9-1), anti-CD69-APC (clone FN50), anti-CD38-PE (clone HIT2), anti-ICOS (CD278)-PE (clone DX29), anti-CD141-PE (clone 1A4), anti-CD66b-PE (clone G10F5), anti-CD137 (4-1BB)-PE (clone 4B4-1), anti-HLA-DR-PE-Cy7 (clone G46-6), anti-CD19-PE-Cy7 (clone SJ25C1), anti-CD25-PE-Cy7 (clone 2A3), anti-CD45RA-PE-Cy7 (clone L48), anti-IgM-BV421 (clone G20-127), anti-PD-1 (CD279)-BV421 (clone MIH4), anti-CD303-BV421 (clone V24-785), anti-CD56-BV421 (clone NCAM 16), anti-CCR7-BV421 (clone 2-L1-A) antibodies. For intracellular staining, cells were first stained for surface markers and subsequently fixed and permeabilized using the Transcription Factor Buffer Set (BD Biosciences-Pharmingen, USA), then stained with anti-Ki-67-BV421 (clone B56), anti-Perforin-Alexa 647 (clone δG9), and anti-Granzyme B-BV421 (clone GB11) antibodies. Following in vitro stimulation assays with specific peptides, cells were stained with the BD Horizon™ Fixable Viability Stain 510 and

anti-CD3-PE-Cy7 (clone SK7), anti-CD4-PerCP-Cy5.5 (clone RPA-T4), anti-CD8-APC-H7 (clone SK1), anti-CCR7-BV421 (clone 2-L1-A), and subsequently fixed and permeabilized using the Cytofix/Cytoperm kit (BD Biosciences-Pharmingen, USA), then stained with anti-IFNγ-FITC (clone 4 S.B3), anti-TNF (clone MAb11) and anti-IL-17-PE (clone SCPL1362) antibodies. Antibody dilutions are available in supplementary Table 2. All samples were analyzed using BD Biosciences - FACSCanto II and FlowJo 10.7.1 software.

**Th1/Th2/Th17 cytokine measurements.** The concentration of seven cytokines, IFN-γ, TNF, IL-2, IL-4, IL-6, IL-10, and IL-17A in serum were measured using a commercial BD CBA Human Th1/Th2/Th17 cytokines kit (cat 560484, BD Inc., USA). Briefly, a mixture of seven capture beads (with distinct fluorescent intensities) coated with capture antibodies specific for each cytokine and a phycoerythrin (PE) detection reagent were used as the manufacturer's instruction. Then, samples (plasma were heat-inactivated at 56 °C for 60 min) were acquired and measured on the BD FACS Canto II flow cytometer and analyzed by FCAP Array software 3.0. Individual cytokine concentrations were indicated by their fluorescent intensities. The value of each cytokine was normalized by the limit of detection.

**In vitro T cells stimulation assays.** PBMC were thawed, assayed for viability, counted, and plated in 96-well plates at $3 \times 10^6$ PBMCs/mL, 100 μL/well in RPMI1640 medium (Sigma-Aldrich - R8758) supplemented with 10% FBS (100 IU of penicillin/mL, 100 μg of streptomycin/mL (Lonza, Belgium) and 2 mM L-glutamine (Lonza, Belgium) (R10H medium), and subsequently stimulated with peptide PepTivator SARS-CoV-2 Prot S (130-126-700 - Miltenyi Biotec, Germany), PepTivator SARS-CoV-2 Prot N (130-126-698 - Miltenyi Biotec, Germany) and PepTivator SARS-CoV-2 Prot M (130-126-702 - Miltenyi Biotec, Germany) at 1 μg/mL. Phorbol 12-myristate 13-acetate (PMA/50 ng/mL, Sigma, USA) plus ionomycin (1 μg/mL, Cayman chemical company, USA) and DMSO were used as positive and negative controls for stimulation, respectively. Stimulation with a cytomegalovirus (CMV) peptide pool at 2 μg/mL (Mabtech, Sweden) was also performed, as a positive control for the assay. All treatments were submitted for 18 h at 37 °C and 5% $CO_2$. 3 h before harvesting, Golgi Plug (BD Biosciences, USA) 1 μg/mL was added to each well. Cells were stained and analyzed for phenotype as described above.

**Enzyme-linked immunosorbent assay (ELISA).** Plasma was tested for IgG and IgA antibodies to S-RBD protein (#RP-87678 - Invitrogen, USA) and N protein (kindly provided by Dr. Ricardo Gazinelli - Fiocruz Belo Horizonte, Brazil) using a protocol described in[17]. Briefly, ELISA plates (Kasvi, Brazil) were coated overnight with 1 μg/mL of SARS-CoV-2 Spike Protein (S-RBD; Catalog nr. RP-87678 - Invitrogen). On the following day, plates were blocked for 1 h at room temperature with blocking buffer (3% skim milk powder in phosphate-buffered saline (PBS) containing 0.05% Tween-20). Plasma samples were heat-inactivated at 56 °C for 60 min and then serially diluted in 1% milk in 0.05% PBS-Tween 20 starting at a 1:25. Plasma was incubated for 2 h at 37 °C. Secondary antibodies were diluted in 0.05% PBS-Tween and incubated for 1 h at room temperature. For both IgG, anti-human peroxidase produced in rabbit (#IC-1H01 - Rhea Biotec, Brazil), and IgA, anti-human peroxidase produced in goat (#A18781 - Invitrogen, USA), was used at a 1:10,000 dilution. The assay was developed with TMB Elisa Substrate - High Sensitivity (Abcam, United Kingdom) for 30 min, and the reaction stopped with 1 M chloric acid. Readings were performed in an ELISA reader (Biochrom EZ 400), and O.D. at 450 nm was used to calculate the area under the curve (AUC), using a baseline of 0.07 for peak calculation[18].

**Statistics and systems analysis.** Percentages were used to describe categorical variables. Pearson's Chi-square test was used to evaluate proportions among children, AMD and ASD. Data normality assumptions were verified for continuous variables and summarized in terms of the median and interquartile range (IQR). A two-tailed Kruskal-Wallis (KW) test followed by Benjamini-Hochberg correction for multiple comparisons was used to compare p-values among the groups. Principal Component Analysis (PCA) was employed to reduce the dimensions of 78 immunological variables generated by flow cytometry analysis, to explain the total variability with a smaller, new set of variables. Spearman correlations were performed between all variables (every set of two variables) and within sets of variables to identify clusters of correlated variables. In PCA of the variables grouped in clusters, variables containing redundant information were excluded. Comparison among groups regarding single variables or PC values was performed by non-parametric Kruskal-Wallis test, followed by 2 by 2 multiple comparisons with p-values adjusted accordingly. Analyses were performed either in GraphPad Prism v.9 or R and sometimes confirmed in Python. Hierarchical clustering analysis was performed using R and a dendrogram was created for the COVID-19 patients using Canberra as the distance measure and Ward's (ward.D) as agglomeration method. The heights used to merge clusters were manually defined.

**Reporting summary**. Further information on research design is available in the Nature Research Reporting Summary linked to this article.

## Data availability

The data supporting the findings of this study are available within the article and its Supplementary Information files or from the corresponding authors on reasonable request. Source data are provided with this paper.

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

## Acknowledgements

Funding for this study was provided by PROADI - HMV, and the Ministry of Health; fellowships for Karina Lima, Julia Fontoura, Renato Stein, and Cristina Bonorino are from CNPq; fellowships for Gabriel Hilario, Priscila Oliveira de Souza, and Tiago Fazolo are from CAPES. TJB is a recipient of an American Heart Association fellowship grant. We wish to thank Drs. André Báfica, Daniel Mansur, Leo Riella, Graham Pawelec and Steve Hedrick for critical readings of this manuscript. Marli Terezinha Crispim for assistance in figure editing. Caroline Nespolo de David for assistance in project management, and the COVIDa consortium (A full list of consortium members appears in Supplementary Note 1). Finally, we wish to thank all the patients who accepted to enroll in the study and donate blood.

## Author contributions

T.F., K.L., P.O.S. and J.C.F. all designed and performed experiments, analyzed data, wrote and edited the manuscript. G.H., R.Z., V.N.P., J.C.F.N., A.F.H., I.G., A.C.O. and R.S.M. assisted in experiments. M.B.B.G performed computational analysis. R.B.G assisted in flow cytometry analysis and interpret the results. L.D.C, A.B and R.E.M. helped with data collection. S.M.C.J coordinated all statistical analysis; I.T.S.S. helped in statistical analysis. L.R.J. helped design the study and edited the manuscript. G.O.Z, I.R.F, F.H.V and M.P.B. helped collect patients' samples and performed RT-PCR analysis. C.S.G. helped collect patients' samples. H.I.N performed clustering analysis. T.J.B. helped design the study, analyzed data and edited the manuscript. M.S.C.C, J.A.S, assisted with epidemiological information. M.C.S. and R.T.S. coordinated the cohort, conceived and designed the study, wrote and edited the manuscript. C.B., conceived and designed the study, performed data analysis, performed principal components analysis, wrote and edited the manuscript.

## Competing interests

The authors declare no competing interests.
