## [Peer Review File · Nature Communications]

Pediatric COVID-19 patients in South Brazil show abundant viral mRNA and strong specific anti-viral responsesREVIEWER COMMENTS

Reviewer #1 (Remarks to the Author):

In this paper, Fazolo et al. performed immune profiling of pediatric and adult COVID-19 patients in Brazil. My review is on the systems biology side of the work.

On page , lines 251-255, the author claims "This approach indicated that the three groups (pediatric patients; mild adult patients -AMD; and severe adult patients - ASD) separated from each other based on three of the new variables created, PC1, PC2, and PC3." Unless we are looking at differ plot, I don't think this is the case. Fig 1B 1C and 1D clearly show that the three groups overlap with each other. Despite the box-plots and associated p-values, there is no adjustment for multiple testing, and I seriously doubt there is statistical significance with the author's claim. I think it is still very likely that the three groups are indeed different, it is just that PC1-3 are not the right features to be used to separate them. The authors can try PC4, PC5, ... or alternative methods of PCA. This is because PCA is restricted to linear transformation, which makes it less effectively to identify the true underlying biological signal. Some other feature selection methods may be needed to achieve dimension reduction needed here.

Reviewer #2 (Remarks to the Author):

The article "Strong specific anti-viral responses in pediatric COVID-19 patients in South Brazil" is very well written, easy to follow and clear. It is a generally descriptive study using a convenience sample of pediatric and adult COVID-19 diagnosed individuals to understand differences in cellular and humoral immune activation at time of PCR diagnosis between children and adults with severe vs mild disease. The statistical approach and analysis is especially clearly written and all the data provided.

Major Comments: none

Minor Comments:

1. Are there any pediatric studies on cytokine/metabolomics/proteomics profiling during acute phase in mild/outpatient disease that you could add to your discussion regarding hypotheses for the differential DC and CD8+ T cell responses you find in the pediatric group?
2. Can you please expand on your finding of low expression of CX3CR1 on the DCs of the pediatric population? Is there a potential hypothesis here as to why this homing signal is under-expressed in milder/pediatric infection? And how could this lead to a more protective response?
3. Line 467 sentence is awkward: "was not determinant for protection" is unclear, maybe change to "did not correlate with protection"??
4. Is there any cohort or previous literature on similar work done in acute RSV or Influenza in a pediatric cohort? This would be very interesting to add to discussion as a comparison in B and T cell responses in those acute respiratory infections that seem to be more severe than COVID-19, you would expect the immune profile/activation to be different.
5. Is there some way to test the hypothesis that pre-existing N-protein specific T cells may be responsible for the lower severity of disease? Are there higher levels in SARS-CoV-2 naïve children than adults in general?

It will be very interesting to see since this study is ongoing, what the next two timepoints will show

and if the responses will continue to be divergent from adults.

Reviewer #3 (Remarks to the Author):

In their manuscript Fazolo and colleagues characterize the immune response to SARS-CoV-2 infection in children, in mild adult cases and severe adult cases. The manuscript is descriptive but probably a good resource for other working on similar questions. However, there are several points that need the authors' attention.

Major points

- 1) Recent papers (e.g. <https://www.nature.com/articles/s41587-021-01037-9>) show that IFN plays a major role in the protection in children. This should be discussed in more detail. Also, the authors do not measure any IFN/cytokines/chemokines in nasal washes or blood – this would have complemented their work.
- 2) In general, this is a very small study and it is descriptive only and these caveats needs to be emphasized.
- 3) Throughout, the language needs improvement.
- 4) The discussion of the data in terms of PCA makes the results section very abstract. It makes sense to perform PCA analysis to see if groups separate. But basing basically the whole manuscript on how groups separate in PCA is not helpful.
- 5) The discussion is overly long and can easily be reduced by half.

Minor points

- 1) Many abbreviations are not defined in this manuscript.
- 2) Line 59: Is really SARS meant (severe acute respiratory syndrome) or COVID-19?
- 3) Line 60 and below: Please specify what is meant by mortality? CFR?
- 4) Line 62: 'COVID-19', not 'covid-19'
- 5) Line 73: It is not clear if children are the main reservoir for these viruses. Studies in e.g. college students show wide circulation.
- 6) Line 75: COVID-19 is the disease, only the virus (SARS-CoV-2) can cause infections.
- 7) Lines 200-202 and other areas: It is unclear why words start with capitalized letters mid-sentence.
- 8) Line 216: What are 'severe and mild adults'? Language!
- 9) Line 360: 'dengue', not 'Dengue'

Reviewer #1 (Remarks to the Author):

In this paper, Fazolo et al. performed immune profiling of pediatric and adult COVID-19 patients in Brazil. My review is on the systems biology side of the work.

On page, lines 251-255, the author claims “This approach indicated that the three groups (pediatric patients; mild adult patients -AMD; and severe adult patients - ASD) separated from each other based on three of the new variables created, PC1, PC2, and PC3.” Unless we are looking at differ plots, I don’t think this is the case. Fig 1B 1C and 1D clearly show that the three groups overlap with each other.

Answer: We agree with the reviewer that there is overlap among the different groups. Some of the individuals from each group most likely have similar cell frequency profiles, and thus appeared close to each other in the PCA graph. However, the focus of our manuscript was not to claim a complete separation of the three groups, but rather to find evidence of average differences and/or similarities in immune responses between children and adults with different clinical presentations of the disease. We have now adjusted the description of the results in the text, rewriting the whole section accordingly, and have also highlighted these changes (starting on page 11, lines 265-272). We hope this will satisfy the reviewer regarding this point.

Despite the box-plots and associated p-values, there is no adjustment for multiple testing, and I seriously doubt there is statistical significance with the author’s claim.

Answer: In the Methods section, we had stated that the p-values of the KW tests are in fact corrected for multiple comparisons by nonparametric post-hoc tests (page 10, line 229-230, now highlighted). If it is found that another level of correction is needed in the new figure 1 because two comparisons among groups were done, one for each PC score (therefore, by multiplying the corresponding p-values by two), the results do not change.

The same rationale is valid if we were to correct for all 18 tests in the old Figure 2, from a total of 25 p-values that are statistically significant, 15 would remain significant and 10 would lose significance. Importantly, the differences between children (the focus of our paper) and the other two groups would remain significant. Because we have now removed the presentation of the PC3 results, the new Figure 2 has 12 tests, if we were to multiply all p-values by 12, the differences for the children will still be significant.

We did debate not presenting all the KW tests in Figure 2, which would eliminate the need to correct p-values for all tests. However, we believe the readers would like to see these results, so we kept them. This is a challenge when you construct a paper integrating more than one field – and that still has no obvious

solution, because as important as the statistics are to validate the conclusions, so are taking notice of tendencies suggested by the data. Once again, this is a first, descriptive study of immune responses comparing children and adults, and differences are as important to us as similarities in this first analysis of the data.

I think it is still very likely that the three groups are indeed different, it is just that PC1-3 are not the right features to be used to separate them. The authors can try PC4, PC5, ... or alternative methods of PCA.

*Answer: The reason for using the first 3 PCs was that PCs are hierarchical – the first ones explain the highest percentages of variance. Using other PCs would imply a substantial loss of information about the data. As suggested, we have now performed a non-PCA method (hierarchical clustering) as an alternative (please refer to the answer in the next point) and opted to include this new result in the **new Figure 1**, substituting the PC3 data. PC3 graphs and loadings have thus been removed from Figure 1 and Supplementary Table 1. We hope the reviewer will approve this substitution.*

This is because PCA is restricted to linear transformation, which makes it less effective to identify the true underlying biological signal. Some other feature selection methods may be needed to achieve dimension reduction.

*Answer: As stated above, we agree with the reviewer that the analysis could benefit from an alternative to PCA that does not perform a linear transformation. We have thus, as requested, performed a hierarchical cluster analysis - which does not perform dimensionality reduction and uses all the information available. The results can be seen in the **new Figure 1**.*

*Using the cell frequencies and gMFI of markers as variables, we performed a hierarchical clustering analysis of the COVID19 patients. The dendrogram was created using canberra as distance measure method and ward.D as agglomeration method. The heights used to merge clusters were manually defined and **added to the methods section, page 10, line 239-242, and to the new Figure 1E**. We can see that the three resulting clusters **are mainly composed of either adults with mild disease, children, or mainly of adults with severe disease. Although, there is again degree of overlap. We believe this alternative analysis complements and agrees with the PCA, bringing insights to the understanding of the disease and how it relates to the immune response of the host. We hope this analysis will satisfy the reviewer's demand.***

Reviewer #2 (Remarks to the Author):

The article "Strong specific anti-viral responses in pediatric COVID-19 patients in South Brazil" is very well written, easy to follow and clear. It is a generally descriptive study using a convenience sample of pediatric and adult COVID-19 diagnosed individuals to understand differences in cellular and humoral immune activation at time of PCR diagnosis between children and adults with severe vs mild disease. The statistical approach and analysis are especially clearly written, and all the data provided.

Answer: We greatly thank the reviewer for these comments.

Major Comments: none

Minor Comments:

1. Are there any pediatric studies on cytokine/metabolomics/proteomics profiling during acute phase in mild/outpatient disease that you could add to your discussion regarding hypotheses for the differential DC and CD8+ T cell responses you find in the pediatric group?

Answer: This article by Loske et al. <https://www.nature.com/articles/s41587-021-01037-9>, published recently, brought up also by suggestion of reviewer #3, is now included in our discussion (pages 19-20, lines 475-483). It helps us answer two points raised independently by the two reviewers.

That article shows results of a single-cell transcriptomic analysis of upper-airway samples from children and adults with COVID-19. They interpret their findings as a pre-activated state of the antiviral innate immune cells. In these samples, they found DCs and macrophages expressing higher levels of MDA5 and RIG I, two pattern recognition receptors relevant for antiviral responses. They have also found that a KLRG1-expressing population of CD8+T cells is higher in children. Both data concur with the two patterns of immune responses we found in the plasma of children in our study, supporting a role for such populations in the differential, possibly protective, immune response in children that is likely associated with milder disease.

2. Can you please expand on your finding of low expression of CX3CR1 on the DCs of the pediatric population? Is there a potential hypothesis here as to why this homing signal is under-expressed in milder/pediatric infection? And how could this lead to a more protective response?

Answer: CX3CR1 expression regulation is still largely uncharacterized, in different cell populations (microglia, monocytes, renal epithelia). It is difficult at present to hypothesize why it would be lower in children. Our finding that it correlates inversely with HLA-DR expression in our samples suggests that children show an activated DC population in the blood that is not migrating to inflamed tissues - and our hypothesis is that this population is protective in favoring antiviral immune responses. We have now rewritten this section in the text, trying to make it clearer in the discussion (pages 20-21, lines 471-483).

3. Line 467 (now, line 492) sentence is awkward: “was not determinant for protection” is unclear, maybe change to “did not correlate with protection”??

Answer: we have corrected the sentence as suggested, and it is now highlighted in the text.

4. Is there any cohort or previous literature on similar work done in acute RSV or Influenza in a pediatric cohort? This would be very interesting to add to discussion as a comparison in B and T cell responses in those acute respiratory infections that seem to be more severe than COVID-19, you would expect the immune profile/activation to be different.

Answer: This Science Translational Medicine study from Heinonen et al., 2020 (DOI: [10.1126/scitranslmed.aaw0268](https://doi.org/10.1126/scitranslmed.aaw0268)) integrated blood transcriptomics and cell profiling with clinical data in a

cohort of 190 children below age 2 years, infected with RSV. In their study, severe disease was associated with low type-I interferon titers and low expression of HLA-DR in monocytes. The interferon association with protection against severe disease in COVID19 seems to be now well established and independently corroborated by other articles, including the Loske et al. paper cited (and now included in this edited version). We have added the Heinonen study to our discussion (page 19, lines 469-471) and reference list, and we thank the reviewer for this suggestion, which we think results in improvement.

5. Is there some way to test the hypothesis that pre-existing N-protein specific T cells may be responsible for the lower severity of disease? Are there higher levels in SARS-CoV-2 naïve children than adults in general?

Answer: The one way to test this would be to have the pre-pandemic, cryopreserved samples of cells from the children, or OF children. However, all individuals were only enrolled after the pandemic had started. We are and all our collaborators are not aware of any group in Brazil that would have such samples.

It will be very interesting to see since this study is ongoing, what the next two timepoints will show and if the responses will continue to be divergent from adults.

Answer: We thank the reviewer for this comment. We are indeed preparing a second manuscript based on the analyses of the memory responses. In our preliminary results, we were able to detect interesting trends for immune memory in children vs. adults. What is most surprising is how fast the children's responses decay over time compared to adults. We hope to submit these novel findings in the next month or so.

Reviewer #3 (Remarks to the Author):

In their manuscript Fazolo and colleagues characterize the immune response to SARS-CoV-2 infection in children, in mild adult cases and severe adult cases. The manuscript is descriptive but probably a good resource for others working on similar questions. However, there are several points that need the authors' attention.

Major points

1) Recent papers (e.g. <https://www.nature.com/articles/s41587-021-01037-9>) show that IFN plays a major role in the protection of children. This should be discussed in more detail. Also, the authors do not measure any IFN/cytokines/chemokines in nasal washes or blood – this would have complemented their work.

Answer: We thank the reviewer for bringing to discussion this recently published article, which had not yet been out at the time of our submission to Nat Comms. Besides the cytokine data in the airways, it also brings some more interesting data - including a minor point raised by reviewer #2 (above). We have now added it to our references, citing it in the discussion (pages 19-20, lines 476-483).

As stated, above, in relation to the cytokine analysis this Loske article, although done exclusively in samples from nasal epithelia, corroborates many of our findings of association of type I interferon and protection from severe COVID-19. We have measured cytokines, and only got the results after we had submitted our manuscript for publication. Our novel findings of plasma cytokines are now summarized in the figure below:

We did not add this to this revised version because: i) we did not measure type I interferon; and ii) the results indicate that only subjects with severe disease showed, at least on average, detectable cytokines in blood. This is a largely expected finding because cytokines are not usually found in blood during mild disease. Thus, we felt this data did not add much new information to the study, other than showing higher amounts of IL-4 (a Th2 cytokine) in the plasma of children compared to AMD – but not ASD. Overall, all cytokine levels were low, and we do not know which cells are responsible for secreting them. We focused our investigation of cytokine production on the experiments with specific CD4+ and CD8+ T cells (Figure 5) – which bring important information on the type of memory responses being generated against the virus.

If the reviewer thinks this graph should be added to the revised version, we could include it as a supplementary figure.

We also think the findings by Loske et al. showing DCs and macrophages expressing higher levels of MDA5 and RIG I - two pattern recognition receptors relevant for antiviral responses - and a KLRG1-

expressing population of cytotoxic T cells that is higher in children, support the distinct immune responses we have detected in the plasma of children which could at least partly explain protection in children. Namely, the activated HLA-DR high, CX3CR1 low DCs; and the TNF+ CD8+ T cells upon restimulation with M and N proteins. We have incorporated this in our discussion, pages 19-20, lines 476-483.

2) In general, this is a very small study, and it is descriptive only and these caveats need to be emphasized.

Answer: We agree with the reviewer. We had already pointed out in the discussion that one of the main limitations of our study was the fact that this is a descriptive study (page 21, line 523). We have now added the limitation of the small sample size, in a sentence that immediately follows (page 21, lines 524-525), and it is highlighted in the text.

3) Throughout, the language needs improvement.

Answer: We have performed a thorough review of the manuscript, and we hope the reviewer will be satisfied with the results.

4) The discussion of the data in terms of PCA makes the results section very abstract. It makes sense to perform PCA analysis to see if groups separate. But basing basically the whole manuscript on how groups separate in PCA is not helpful.

Answer: We have tried to make the PCA interpretation less abstract by summarizing possible biological interpretations and predictions as we wrote each paragraph (lines 292-301; 304-307; 353-357; 361-362 and 378-383). PCA analysis is restricted to the first three figures of the article, dealing with the analysis the non-specific immune responses.

5) The discussion is overly long and can easily be reduced by half.

Answer: We have now edited and shortened the discussion.

Minor points

1) Many abbreviations are not defined in this manuscript.

Answer: Abbreviations have been revised and corrected. They are now highlighted throughout the text.

2) Line 59: Is SARS really meant (severe acute respiratory syndrome) or COVID-19?

Answer: We have now corrected on page 3, lines 60-61: "SARS by" has been deleted. We thank the reviewer for pointing that out

3) Line 60 (now, 66) and below: Please specify what is meant by mortality? CFR?

Answer: Yes – CFR- estimates of the proportion of deaths among identified confirmed cases. We have included this information on line 66, now highlighted.

4) Line 62 (now, 69): ‘COVID-19’, not ‘covid-19’

Answer: Spelling was corrected and highlighted in the main text (page 3, line 69).

5) Line 73 (now, 80): It is not clear if children are the main reservoir for these viruses. Studies in e.g. college students show wide circulation.

Answer: We agree. We have re-written the sentence, including this reference by Davis et al. 2018 Sep;12(5):582-590. doi: 10.1111/irv.12563, reporting circulation of seasonal coronaviruses in college students – page 4, lines 80-81.

6) Line 75 (now, 83): COVID-19 is the disease, only the virus (SARS-CoV-2) can cause infections.

Answer: We thank the reviewer for pointing that out. It has now been corrected and highlighted in the text, now line 83: “...after infection by SARS-CoV-2.”

7) Lines 200-202 (now, 210-212) and other areas: It is unclear why words start with capitalized letters mid-sentence.

Answer: Now on line 210, page 10 it is the commercial name of the product, thus capitalized: Spike Protein (S-RBD; Catalog nr. RP-87678 - Invitrogen). Now line 212, we have now corrected the capitalized words and highlighted them in the text: ... (3% skim milk powder in phosphate buffered saline (PBS)).”

8) Line 216: What are ‘severe and mild adults? Language!

Answer: We thank the reviewer for pointing that out. It has now been corrected and highlighted in the text, now lines 226-227: “Pearson's Chi-square test was used to evaluate proportions among children, AMD and ASD.”

9) Line 360: ‘dengue’, not ‘Dengue’

Answer: Spelling was corrected and highlighted in the main text page 15, line 374.

REVIEWERS' COMMENTS

Reviewer #1 (Remarks to the Author):

The authors are responsive to my comments. I am satisfied with most of the answers. Just one thing, the authors stated that "Using other PCs would imply a substantial loss of information about the data." This may be correct but the point is that the true signal may be weak and subtle, especially compared to non-biological signals such as batch effect or experimental artifacts. So it is common that the first one or two PCs are skippd for not being biological meaningful. It is possible that PCs3-5 are more informative for clustering.

Reviewer #2 (Remarks to the Author):

The author's responses and edits to the reviews are sufficient for publication, I have no new comments/changes to request.

Reviewer #3 (Remarks to the Author):

The authors did address my comments well.

Reviewer # 1(Remarks to the Author):

“The authors are responsive to my comments. I am satisfied with most of the answers.

Answer: We are happy to read that. We thank the reviewer for all the comments and the suggestions.

Just one thing, the authors stated that "Using other PCs would imply a substantial loss of information about the data." This may be correct, but the point is that the true signal may be weak and subtle, especially compared to non-biological signals such as batch effect or experimental artifacts.

Answer: We agree with the reviewer, and we believe we have been especially thorough in our analyses to identify relevant biological information

So, it is common that the first one or two PCs are skipped for not being biological meaningful. It is possible that PCs3-5 are more informative for clustering.

Answer: We appreciate the possibility raised by the reviewer. So, we have performed the analysis suggested, and constructed graphics using these principal components. PC3 separates children from adults with severe disease – with still some overlap. However, this result had already been shown by PC1, both in the former Figure 1 (original version); and the current Figure 1B (revised version). When we plot PC 4-5, there is no separation of the groups. Thus, we do not believe the possibility raised by the reviewer applies in this set of data and PC 3-5 do not seem to add new information about the differences among patient groups. PC1 and 2 alone explain the highest amount of variance and show separation (with some overlap) of the groups, which is confirmed by the hierarchical clustering analysis (Figure 1E in the previously submitted revised version).

We have now included the analysis of PC3-5 as Supplementary Figure 2A; and corrected the text, accordingly, including this information, on page 6, lines 132-137. We hope this additional analysis will satisfy the reviewer.

Reviewer #2 (Remarks to the Author):

The author's responses and edits to the reviews are sufficient for publication, I have no new comments/changes to request.

Answer: We are happy to read that. We thank the reviewer for all the comments and the suggestions.

Reviewer #3 (Remarks to the Author):

The authors did address my comments well.

Answer: We are happy to read that. We thank the reviewer for all the comments and the suggestions. The cytokine analysis was included as Supplementary Figure 4B.